# Guided Electromagnetic Wave Technique for IC Authentication

**DOI:** 10.3390/s20072041

**Published:** 2020-04-05

**Authors:** Mosabbah Mushir Ahmed, Etienne Perret, David Hely, Romain Siragusa, Nicolas Barbot

**Affiliations:** 1ORSYS & CTSYS, LCIS, University of Grenoble Alpes, 26000 Valence, France; Etienne.Perret@lcis.grenoble-inp.fr (E.P.); david.hely@lcis.grenoble-inp.fr (D.H.); romain.siragusa@lcis.grenoble-inp.fr (R.S.); nicolas.barbot@lcis.grenoble-inp.fr (N.B.); 2Institut Universitaire de France, 75231 Paris, France

**Keywords:** counterfeit IC, EM, FPGA, process variation

## Abstract

Counterfeiting of an Integrated Circuit (IC) has become a significant concern for electronics manufacturers, system integrators, and end users. It is necessary to find a robust implementation that is efficient, low cost, and noninvasive in detection and avoidance of ICs counterfeiting. In this paper, we introduce the concept of using a guided radiofrequency (RF) wave technique to authenticate ICs. The approach discussed in this work highlights the use of electromagnetic (EM)/radiofrequency (RF) response that has been further evaluated to assign fingerprint or signature of ICs for the purpose of authentication. Our approach is to use EM/RF guided wave to sense the response of the ICs, extract the manufacturing-based process variation of an IC and finally generate identifier or signature of that IC. As a proof-of-concept, we performed experiments over different field-programmable gate array (FPGA) boards of the same family. The post-processing technique was applied on the measurement results to statistically quantify the error probability of the authentication technique.

## 1. Introduction

The current economy largely depends upon the growth of electronics and semiconductor devices. An Integrated Circuit (IC) can be found in various domains, from basic kitchen appliances to critical applications, like aerospace, military, healthcare, etc. Therefore, ICs can be termed as a root of system trust [1,2]. Owing to reliance of ICs in various applications, it is important to ensure their authenticity as their failure can lead to disastrous consequences. Along with the multi-faceted use of ICs in various domains, there are always issues regarding the shrinking size of transistors and other devices. Hence, it has become apparent that designers and manufacturers face multiple challenges when it comes to employing strategies to tackle problems related to IC authenticity. In a practical scenario, such as internet-of-things (IoT) or resource constraint environments, a manufacturer or designer needs to be sure that they are using a trusted or genuine IC in their product [3,4,5].

Classically, authentication is set of steps that guarantee a device in use is genuine. Among various techniques of authentication, some approaches involve utilization of stored pins or keys, passwords, bar-code on devices, etc. Another method is based on the principal of harnessing the natural randomness of the device or entity. A prime example of such randomness-based authentication step is utilization of human bio-metric fingerprint and retina. Such techniques are robust against cloning and reverse engineering types of attack [6,7]. To adopt the bio-metrics for authentication classically from Reference [8,9], sensors are utilized that deploy a transducer that changes a bio-metric trait (like fingerprints, iris, etc.) of a person into an electrical signal. Similarly, in this paper, we devised a methodology that can sense the inherent randomness of the ICs and create fingerprints or signatures for them. 

The idea of implementing this study was to introduce a novel and noninvasive technique for authentication of ICs. While ICs can be reliably mass manufactured to have identical functionality, the premise of our approach is that each IC is unique in its physical characteristics due to the inherent variations in manufacturing across different dies, wafers, and processes. Our work proposes to exploit natural randomness of ICs and enhance its effect through an electromagnetic (EM)/radiofrequency (RF)-based approach. To the best of our knowledge, this is the first time this method has been used for the authentication purpose of the ICs. 

The principle idea of this work is utilization of the RF/EM waves-based technique, wherein a guided RF wave is made to interact with the internal physical structure of IC, and the obtained response of interaction is used to generate an identifier that is applied for the authentication purpose. The central idea of this study is to exploit the underlying process variation (PV) effects through an injected guided wave. The underlying principle of exploiting the elemental natural randomness or manufacturing vulnerabilities of the IC is similar to extracting and sensing a human fingerprint or iris pattern for the authentication. A high level depiction of the idea is shown in Figure 1. As observed in Figure 1, a set/batch of ICs are treated (by user or manufacturers) by guided RF input, and their obtained RF responses are recorded. Observing Figure 1, due to the fabrication variation, different ICs excited with the same input signal would generate different output signal that can be used for authentication. The depiction in Figure 1 is just for illustration purposes; it is not the exact signal at input and output. In addition, taking Figure 1 as a reference, we can observe that the whole authentication and identifier generation also comprises of a post-processing or mathematical treatment part. Once the RF responses are obtained, it is essential to apply some mathematical treatments in order to quantify each response distinctively. Going forward in this work, we highlight the usability and implementation of quantitative analysis of the obtained response from our measurement steps. The idea of this study going forward is to setup a system of experimental measures that is robust and efficient enough to exploit the PV effects of an IC in a noninvasive manner by using a guided RF/EM signal. Further, the statistical methods are applied in order to quantify the results in terms of false positives/negatives and error rates.

The rest of this study is divided into understanding the trust issues in IC and classical authentication techniques. Later, the methodology of the guided RF-based authentication technique, responses, and results are introduced. Before understanding the detail of our approach, we briefly highlight the present techniques to authenticate ICs that have been studied. In addition, we discuss the motivation behind developing our novel technique in the present scenario of IC and semiconductor applications and segments.

Subsequently, we elaborate on the details of the test and experimental approaches, different challenges, and how the results were computed to have an effective signature for the purpose of authentication of ICs. In our study, we also took into account the various systematic and printed circuit board (PCB)-related errors, and we designed concepts and experiment to mitigate such an error.

## 2. Counterfeit Detection and Avoidance Techniques

### 2.1. Classical Techniques

Over the past several years, specialized services of testing have been created for detecting and avoiding counterfeit components. The components must be authenticated by these tests before being placed in systems. The classical techniques to detect counterfeit that involve physical and electrical inspection can be very time-consuming and involve the risk of damaging the IC or components under test, permanently or temporarily [10,11].

Physical inspection methods include examination of the components documentation, exterior and interior inspection with low and high power visual inspection equipment, and material analysis of the device under test (DUT) [1,12]. The electrical inspection phase of the tests includes AC/DC parametric tests, functional tests, and burn-in tests. While physical inspection methods are effective in detecting poor quality recycled parts, they are expensive, time-consuming, and destructive, while also being less effective for more sophisticated recycled components. Note that electrical tests can also be very costly [13].

### 2.2. PUF-Based Techniques

The second method is based on traceability approach to find identity of devices or ICs by using manufacturing-based PV effects. As discussed, using PV approach, physical unclonable function (PUF) is dominantly used. It exploits and senses the inherent variability of an IC, caused due to manufacturing variations of the IC itself. Each PUF contains pair of challenge and response. For each challenge sent to an IC, there is a unique response to that challenge [14,15]. Apart from the IC authentication, PUFs are also used for the purpose of secret key generation for cryptographic applications [15].

The bottleneck of PUF approach is that it requires dedicated on-chip circuitry, which may be complex to process, implement, and industrialize. The on-chip post-processing involves a lot of error correction mechanisms, along with basic post-processing circuit elements. Hence, this requires lot of chip area. PUFs also require large database of challenge pair response. Moreover, PUFs have been subjected to model-based attacks; hence, the identity or uniqueness of key can be compromised [16]. The entropy of PUF depends upon the error correction scheme, and a number of challenge response pairs, which results in the loss of information due to that. Apart from the usage of classical techniques and PUF-based solutions, research and studies have been ongoing to develop new approaches that would be effective against counterfeit problems. Even though PUFs tend to have few bottlenecks, they are still the most adopted choice for authenticating the ICs. Their bottleneck does not completely diminish their usage in various applications of hardware security.

## 3. Guided EM Wave Principle

As discussed previously, classical testing techniques can be ineffective in some instances and pose risk to damage ICs. With the miniaturization of IC, there are various constraints that arise to add new circuit elements in the IC. Area overhead is a major constraint for semiconductor vendors and designers. Most widely-adopted techniques, like PUF, also suffer from the area overhead constraint. Hence, adding any extra dedicated circuitry in an IC for the purpose of authentication can be discouraging in terms of economic viability for any manufacturer, particularly for small and resource constraint devices. Therefore, it is important to search for a solution that considers the area constraints of IC, while implementing solutions to address the issue of counterfeiting. In this study, the proposed scheme is aimed to provide a solution that is noninvasive in nature, efficient to check for authentication, risk-free, and able to be applied across different type of semiconductor devices (ICs). 

### 3.1. Implementation Overview

The approach discussed in this study justifies the use of RF/EM technique, i.e., excite the underlying PV effects through the RF waves. The wave is guided into the IC via the transmission line (refer to Figure 1 for high level illustration). The wave is partially transmitted, and a part is be absorbed or reflected back. The response of the transmitted and reflected wave is linked to the architecture or internal physical structure of the IC. The study focuses on utilizing the basic internal manufacturing defects or PV effects (like in human fingerprints, etc.) that come from irregularities of wires/interconnects, transistor mismatch, etc. [17,18]. There is no implementation of any specific type of marker or sensors, as is mostly done in cases of PUFs. The idea is to sense and measure the amount of disruption a guided wave experiences when it interacts with the manufacturing physical variations of the IC. 

Elaborating the illustration of Figure 1, a wave is injected in the IC, and its transmission response named S21 response (relation between input-output) is calculated. The scattering parameter or S-parameter quantifies how RF energy propagates through a multi-port network. The S-matrix is what allows us to accurately describe the properties of incredibly complicated networks as simple “black boxes” [19]. The ICs are not RF devices, so consideration must be taken about how to connect the RF wave to the input of IC. For this reason, the approach rests on the fact that a part of the input RF wave can propagate through the device. In return, the measurement of the relation between the input and the output power can be measured with high performance equipment (vector network analyzer (VNA)) with high dynamics (120 dB); so, even a very small part of the propagated signal at the output can be relevant to generate a fingerprint. Hence, this way, we can prove that enough energy is present that can be used to generate the fingerprints of the IC. 

### 3.2. Authentication Steps

A scheme of authentication is shown in Figure 2. Referencing a previous study [13], the steps to authenticate an IC using this approach can be divided into 2 stages: *enrollment stage* and *verification stage*.

During the *enrollment stage*, a set of RF measurements are performed on IC under test. The obtained RF response is post-processed (mathematical treatment) to generate an identifier or a signature. The illustration in Figure 2a shows the construction of the database. The measurement results are stored in a database after being post-processed (depicted as PP_A_ in Figure 2a) using software computation. The database is used as a reference to check for the results of the IC measurement and verify if they are genuine or not. During the *verification stage,* the IC under test is subjected to the same measurement procedure. The response is post-processed and compared with that of the database.

From Figure 2b, it can be understood that IC during verification stage is subjected to same measurement steps as those conducted during the enrollment stage. The post-processed response from the verification stage, as depicted as PP_B_ in Figure 2b, is compared with the responses stored in the database (obtained from enrollment phase). The decision is made to determine if the signature of both stages is same or not, i.e., IC authentic or not. The next steps, after understanding the basics of the authentication steps, is to comprehend the implementation strategy, like measurement setup, PCB development, etc., prior to the comparison of the signals and analysis of results.

## 4. Measurement Bench

For experimentation and measurement of the proposed technique, in this work, we developed a customized RF-PCB test bench. In this study, we used 11 field-programmable gate arrays (FPGAs) of the same family, manufacturer, etc., as DUT. A similar technique can also be extended for other application specific integrated circuits (ASICs) or analog ICs; however, FPGAs (owing to reprogramming features) are used here to prove the validity of concept. We designed a RF-PCB board for the measurement purpose. To implement such a measurement setup, it is necessary to understand and to evaluate the possibility of integrating an IC (DUT) on a PCB that has the capabilities to work in RF range. The challenge here is to characterize and develop the PCB or test-bench and determine the components for proper transfer of the RF/EM wave though the IC. It is necessary that a considerable part of the RF signal transfers through the IC under test. In order to transmit the maximum power of the incoming RF signal through the IC under test (FPGA), the simplest way done here, is by adding a 50 Ohms resistor parallel with the FPGA; like most of digital circuits, the FPGA has high impedance Input-Output (IO) pins.

A pictorial depiction of the PCB developed for the measurement and implementation purpose is shown in Figure 3. The PCB is developed on FR-4 substrate and is a 4-layered board capable to work up to 2 GHz. As shown on Figure 3, the PCB houses a SPARTAN-3A FPGA (based on 90nm CMOS), SMA connectors (for RF IO) and other auxiliary circuits for power management. For the propagation of the RF signal, 50 Ohms transmission lines that support the proper injection of the RF signals in the FPGA (IC under test) are used. To program the FPGA, the PCB has a JTAG connector. To the best of our knowledge, this is the first time this type of configuration, namely wherein RF signal interacts directly with a FPGA, has been done.

The setup shown in Figure 3a is used to perform the experiments in this study. This setup is used to justify the proof-of-concept. For other types of IC (analog IC or micro-controllers (MCUs)) or in a more industrial scenario, we can opt for another type of test-bench of PCB but with a similar central idea of exploiting the PV effects using guided wave technique. 

In a more industrial scenario, a better plug and test setup can be deployed, as shown in Figure 3b. Understanding the illustration from Figure 3b, an IC or DUT of a particular package type to be authenticated can be plugged in a plug and test kind of PCB set up, with proper IO connectors to support the transfer of RF power.

This technique can facilitate the usage of various ICs of a similar package to be subjected to guided RF waves, and their IO response can be characterized. This characterization can be further extended for the purpose of generating signatures.

For the measurement purpose, we used a vector network analyzer (VNA), a DC bias tee, and a power supply. The VNA is used for the measurement of output responses, like S21 parameters. The DC bias tee is used for conditioning the proper DC voltage to the buffers or logic elements of FPGAs. An illustration of experimental setup is given in Figure 4. The input DC bias for the circuit programmed in FPGA is set at 1.5 V. At DC bias of 1.5 V, a proper operating condition is established that supplies a steady voltage for device to operate. We used 1.5 V as the DC bias, a middle voltage level between 0 and 3.3 V. The 0 V input corresponds to logic level 0 and 3.3 V for logic 1. The RF input power level generated from VNA is set at 10 dBm. The high power is needed because the IO pins can block the RF signal at the input pins of the IC. The selection of the bias voltage, input power level, and other related parameters can vary depending upon the experimental setup, choice of DUT, etc. In this study, we chose the aforementioned parameters, taking in to account the proper biasing level that the digital circuit of this family of FPGA DUT required to have a stable response. In addition, as in Figure 4, the reference plane for calibration purposes using VNA was chosen at the SMA connectors.

## 5. Implementations in FPGA: Exploit PV Effects

### 5.1. FPGA Programmed Circuit

The main aspect in creating the signatures is to exploit the underlying manufacturing-based PV effects of the IC. In this work, we did not opt for any variability aware circuit to accentuate the PV effects has been done in other studies, like Reference [17,18]; rather, our objective was to exploit the PV effects from regular circuit elements, like a basic logic block (configurable logic block in FPGA) and IO routing (interconnect). The FPGAs in this work are programmed with an interconnection between IO ports with only a buffer circuit detailed later. The injected RF wave gets perturbed by the interconnect and buffer circuit it finds in its traversing path.

The interconnects deployed in the FPGA can be defined as a conductive connection between an input and output ports capable of carrying a signal. At low frequencies, a wire or an interconnect track may be an ideal circuit without resistance, capacitance, or inductance. But, at high frequencies, AC circuit characteristics dominate, causing impedance, inductance, and capacitance to become prevalent in the wire [20,21]. The length and width, etc., i.e., physical features, are not the same on two ICs of the same batch, configuration, mask, etc. [6].

In terms of the physical layout, the effects, like line edge roughness (LER), can effectively cause variations in the resistor, inductor, capacitor (RLC) dimensions even if the same exact protocols and layout masks have been used. An equivalent RLC model of the wire is given in Figure 5 for the purpose of highlighting the physical implications of the wire or interconnects. The incoming RF wave can get disturbed by the effects of physical features of interconnect; hence, the response of two similar ICs can be distinguished by only the use of interconnects [13].

In addition, considering the effects from the circuit elements—the non-linear effects from the logic circuit of FPGA, which consist of various RC parasitic on its transistor level modeling, causes considerable perturbations to the incoming RF wave, along with the interconnects. Hence, owing to the PV effects a digital circuit (non-linear transistor element), along with interconnects (see Figure 5), can have a significant impact on the cut-off frequency of the incoming RF wave. This can also be a characteristic cut-off frequency for each DUT or FPGA.

### 5.2. Multiple Routes between IO Pins

For the purpose of experimentation, we programmed all the FPGAs with two different routes—long and short—on the same IO pins. A pictorial depiction of implementation of the routes and buffer circuit (a unit digital circuit) in the FPGA is shown in Figure 6. The illustration can be observed from Figure 6a,b where, between IO ports ‘A’ and ‘D’, there are two different lengths of route through which the RF wave traverses through the FPGA. The routes signify the interconnection between input and output pins through a buffer circuit in their path. All measurements are performed using the results from these two routes. Of course, a higher number of routes can also be established, but here we have focused to keep the implementation simple in order to highlight a proof-of-concept.

During the measurement, at one time, the FPGA in use is programmed with only one of the two routes. For example, consider a FPGA named as ‘*FPGA 1*’. Firstly, it is programmed with only short route, and all its measurements are performed and responses obtained. Once all measurements with short route are completed, ‘*FPGA 1*’ is programmed with long route, and measurement steps are performed and results noted.

### 5.3. Experimental Steps

In order to perform the experiments using the setup of Figure 4 and protocols of Figure 6, we outlined some of the practical steps that are required and were carried out in this study:Select the DUT to be authenticated.Adapt and customize a testbench to carry out the experiments efficiently.Select required instruments and determine various electrical parameters that ascertain to correct operation of the DUT for proper experimentation.Select the technique that can accentuate the exploitation of PV effects.Record the response and post-process it numerically to determine statistical difference between responses of the DUTs.

The aforementioned five steps can be guideline to launch a successful experimentation protocol for robust and reliable measurements. Of course, for different DUTs and experimental situations, the parameters of tests and setup can be varied, but the core steps would remain similar.

## 6. Measurement Results

The setup shown in Figure 4 is used to perform the measurements on 11 FPGAs following the experimental steps discussed in above Section 5.3. The result of the experiments is discussed in this section. The results are divided into two parts. In the first part, we highlighted the S21 curve response for all DUTs for varying lengths of the interconnects. Secondly, in order to measure the statistical distribution of the result, we determined the false positives and negatives. This is done in order to take into account various measurements errors that can affect the results and how to numerically analyze the results.

### 6.1. S21 Parameter Response

For the measurement purpose, the RF wave is guided through the SMA connectors through the transmission line into the FPGA, as also discussed in detail in Figure 4. The DC bias provides proper voltage condition for the IO buffer of the FPGA to conduct the RF wave inside the FPGA.

As also discussed in Section 3, the transmission coefficient or the S21 response is measured in this work to obtain the relation between input and output RF signals. An illustrative depiction is shown in Figure 7, wherein Figure 7a shows the response of full bandwidth used in the measurement (here shown for two FPGAs only to highlight the extent of response when full measurement is used), and Figure 7b depicts the response from two routes concentrated in the region up to a cut off frequency of the FPGAs. The S21 response for the two routes (short and long) for 11 FPGAs with repeated measurements is shown in Figure 7b. 

The S21 parameter is linked to show the injected RF wave is affected when it traverses through the interconnects of FPGA and interacts with the buffer circuit. The result highlights that each FPGA has a distinct cut off frequency for the same RF input wave. This cut off is linked to the internal physical features determined by PV effects.

From Figure 7, each FPGA was measured 10 times with the setup shown in Figure 4, following the exactly same experimental protocols and steps. The graph (in pink color) in the inset in Figure 7b depicts the repeated measurement of one FPGA multiple times to highlight the robustness of the response. The graphs from Figure 7 clearly depict that the S21 curves for 11 FPGAs are different from repeated measurements on both the routes. This difference in S21 response for each FPGA can be utilized to differentiate them and further use it for authentication purpose of the FPGAs.

Furthermore, from Figure 7, it is observed that each FPGA of the same family, programmed with same configuration, gives a different S21 response. The cut off frequency is determined in the same way as that of filter behavior of the buffer and RLC components of the interconnect/routing. This determines that, even if the devices are of the same family and configuration, their manufacturing-based PV effects can be exploited by the guided RF waves that can accentuate a difference in the response from each device. This can effectively be used to generate a signature or fingerprint for the authentication scheme.

### 6.2. Statistical Distribution of the Response

The observable distinctions in the S21 response emphasize the use of guided RF technique to exploit PV effects. However, any measurement can have errors that can affect the repeatability of results. Such fuzzy characteristics are caused by the physical randomness that introduce entity-specific features during manufacturing, which are typically not uniformly distributed. The fuzziness of an RF-based response can be most clearly depicted by its inter- and intro-distributions and computing error probability rate by finding the false rejection rate (FRR) and false acceptance rate (FAR), respectively [15,22]. 

In this work, we opted to use cosine similarity (CS) as a mathematical tool that groups the responses from each measurement on each device into inter- and intra-responses. CS is based on finding the cosine of angle between two datasets; its scores are limited between 0 and 1. If the value is 0, then degree of dissimilarity is high between the datasets; if it is 1, the response from two sets is similar. We performed CS computation on the complex value of the S21 response, wherein both the magnitude and its phase are taken into calculations [23,24]. The RF response, obtained through the measurement as discussed previously, has been subjected to a CS-based computation technique. This technique enables the determination of the rate of error probability. The error probability curve is shown in the Figure 8. From Figure 8a,b, the obtained error rate (computed for every repeated measurement of 11 FPGAs) for the short route is 10^−3^,and, for the long route, it is 10^−4.^ The histogram distribution of the CS score of all measurements of both routes, highlighting inter- and intra-device variability, is depicted in Figure 8c,d. The low values of error probability (for both routes) validate that it is possible to distinguish devices (FPGAs) based on their transmission response characteristics. This can be effective for the purpose of authentication of FPGAs using guided RF waves. Hence, based on the S21 responses, we were able to find a good distinction between two FPGAs, even if they are of the same family, series, or manufacturers.

The results shown in this part highlight the efficacy of using this technique to exploit the PV effects of the ICs. To improve upon the results and take into account the effects of measurement systematic errors, we made a study that determines the mitigation of systematic noise from the measurement setup used in this study. The following subsection is an elaboration of such an effort of systematic noise mitigation.

### 6.3. Mitigating Systematic Errors

On a PCB, the signal propagates not only into the DUT (FPGA in this case) but also in the transmission lines of the PCB. Difference in manufacturing of this line (and soldering effects, etc.) can affect the signal. Hence, we can say that the difference observed previously is also affected by these PCB-based errors [16].

Going forward, we proposed techniques that can be used to mitigate the effects of the systematic errors due to the PCB manufacturing and solder joints, etc. In this work, we deployed a two-routes approach technique in order to mitigate the systematic errors related to the PCB. The basic idea, as depicted in Figure 9, is to change the reference plane of calibration from SMA connectors to FPGA pins. From Figure 9a, the blue-colored rectangle represents a reference plane for the calibration. It is clear from this representation that the calibration plane is the SMA connectors. This setting was considered in the earlier measurement and post-processing cases. To remove the errors, the reference plane needs to be changed from SMA to the FPGA pins, as is shown in Figure 9b. This effort can remove effects of the systematic error from the PCB. Hence, only the variation from the FPGAs (IC) is taken into the results.

### 6.4. Differential Setting for Error Removal

From Figure 9, we described the process that can be undertaken to mitigate the variations coming from the PCB board. The implementation technique for that is done by performing the subtraction of the responses obtained from the two different routes—short route and long routes—as programmed in FPGA.

Computing the subtraction from two routes can be effective in mitigation or removal of the errors coming from PCB and other external sources. A pictorial depiction is shown in Figure 10a, wherein the errors from the two routes can be removed by subtracting the responses. Deploying the same two routes that were used earlier, we can compute the difference in the response from two routes, as shown in Figure 10b. This is equivalent to changing the reference plane of calibration, as proposed in Figure 9. The CS-based mathematical treatment can then be applied on the response obtained from the difference of two routes and error probability, and statistical distribution can be obtained.

Using the strategy described in Figure 10a,b, we deduced the error probability curve of the subtracted response from all the 11 FPGAs under test. From Figure 10c, we can observe the obtained error probability after subtracting the response from two routes. The error rate in this case is obtained in the range of 10^−2^. This error rate is slightly higher than what we obtained in previous cases.

We can refer to Table 1 to observe error probability for various cases. Even though the error probability is higher with the difference approach, with this we can remove the systematic errors coming from PCB. After using the subtraction (differential-based scheme implemented here), only the randomness related to the FPGA comes into consideration during the computation. Therefore, the result is related to all the information linked only to the FPGA.

## 7. Conclusion and Prospects

### 7.1. Inference

We already observed distinction among the S21 responses (from Figure 7) from the 11 FPGAs of the same family, series, etc. In addition, using the CS-based mathematical treatment we find the intra- and inter-distributions and the corresponding error probability. Observing the results from Figure 8, the overlap error or error probability for all the FPGAs were low. In Figure 8a, error probability for short route is observed to be 10^−3^, and, for the long route in Figure 8b, it is around 10^−4^. Such low values of error probability (for both route) validate the usage of guided RF wave to interrogate and exploit the underlying PV effects, which can be useful in the purpose of authentication. 

Observing the results in Figure 7, we also clearly see that each FPGA, despite being of the same family, manufacturer, series, etc., has distinct responses when perturbed with a guided RF/EM wave. In addition, using CS-based computations over repeated measurements on several FPGAs, we can observe that we have low error probability, and inter- and intra-distribution never overlap. Therefore, the techniques—guided wave and CS-based—applied in this work can be an effective way to generate an identity of FPGAs (or ICs) for the purpose of the authentication.

Furthermore, the nuance of implementing the differential setting (Figure 9 and Figure 10) validates the effort that removes the systematic errors from the PCB. The results in terms of error probability from Figure 10c and Table 1 show that even though there is a trade-off in terms of error probability rate, the results are directly linked to the FPGA (or DUT) using this technique. Consequently, we can interpret that, using guided wave technique and applying different schemes to improve the methodology, we are able to exploit the PV effects from the FPGAs. We can say that this is an approach that is highly noninvasive in nature and can be effectively used in different industrial settings. Using such approach, counterfeit techniques, like recycled, remarked, over-produced, etc., can be easily detected and avoided. Of course, with each technique, there are few bottlenecks. We highlighted few limitations of this approach in the section below.

### 7.2. Limitations of the Proposed Technique

Given the novelty of this approach, there can be few aspects that may need to improve in future works to make the methodology more robust and efficient in implementation. For example, the DUT requires a specially customized PCB. Hence, it can be time-consuming and may not be easily adapted to some of the IC package. However, given the advancements in technology, there are many PCB which can allow a plug and play situation (as also discussed in Section 4). So, several ICs of the same package can be embedded on the same PCB, and the measurements can be performed. Hence, this can mitigate the need to develop a customized PCB. 

### 7.3. Conclusions and Future Perspectives

The proposed novel method of a guided wave-based approach was effectively used to generate a signature of each DUT by using very basic circuit elements. The measurement results were calculated, and response was treated with post-processing techniques that numerically determined the response. Taking into account the noninvasive nature of the approach, it shows that there was no stringent requirement to implement a circuit to exploit the PV effects. The obtained signatures were subjected to mathematical treatment to observe the error probability, uniqueness, and robustness parameters. The results show that it is possible to extract signatures that have enough randomness to be unpredictable, unique, and robust. The error probability obtained using CS-based technique has been low. The study conducted determined a proof-of-concept that, by using noninvasive technique, i.e., without the implementation of the dedicated circuit, the PV effects of an IC could be exploited. Moreover, the experimentation steps discussed in this work can be further improved upon by studying the other features of various types of the DUTs. FPGAs here present different sets of challenges, which may not be same when we perform the measurements on other types of ICs. Hence, it will require further investigation. Going forward, we will focus on reducing the error probability further by increasing randomness. In future work, the idea is to increase the number of the FPGAs or DUTs to validate the highlight of this study on many devices. In addition, we will study the steps and mathematical computation of the error removal techniques in detail. Similar steps can also be conducted on analog ICs and other ASIC in the future.

## Figures and Tables

**Figure 1 sensors-20-02041-f001:**
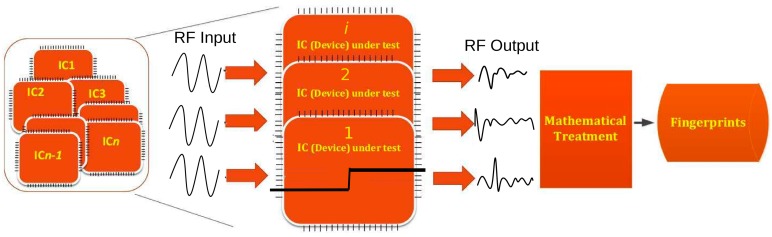
An overview of high level implementation of authentication using guided wave technique. Each Integrated Circuit (IC) taken from the same batch produces distinct radiofrequency (RF) output for the same RF input, which can be post-processed to generate a sort of fingerprint for them.

**Figure 2 sensors-20-02041-f002:**
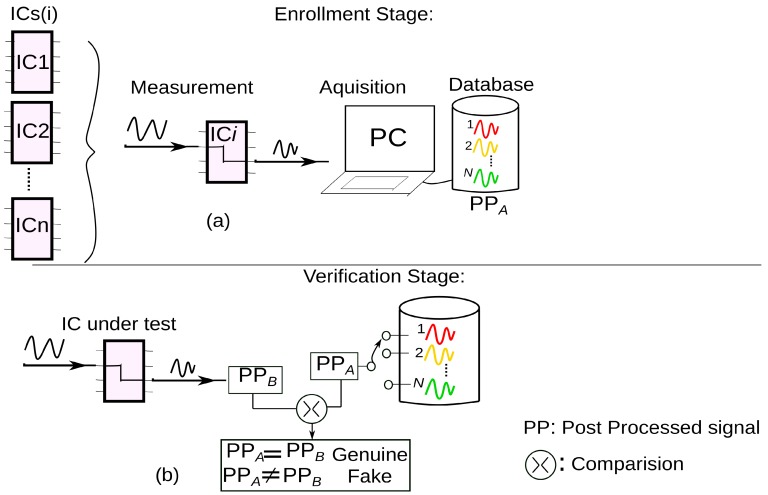
A depiction of different steps involved in the authentication procedure, implemented using guided wave approach. (**a**) Enrollment stage, which generates the database before IC is sent to the end users. (**b**) Verification stage: End users or customers who want to check for authenticity of the ICs they have. PP = post-processed.

**Figure 3 sensors-20-02041-f003:**
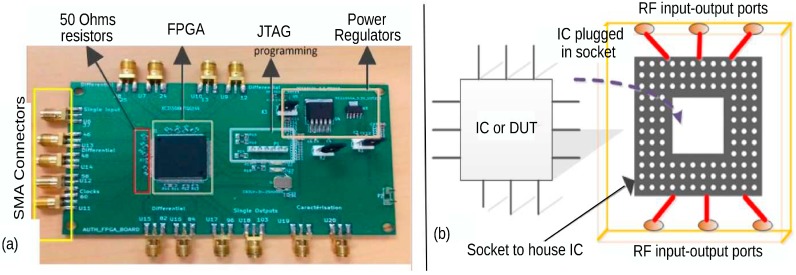
Device under test (DUT) setup. (**a**) A pictorial description of RF-PCB developed on FR-4 housing field-programmable gate array (FPGA) as DUT. It also highlights various axillary circuits and SMA connectors. (**b**) An illustration of using IC on a pluggable socket to be characterized with the guided RF waves.

**Figure 4 sensors-20-02041-f004:**
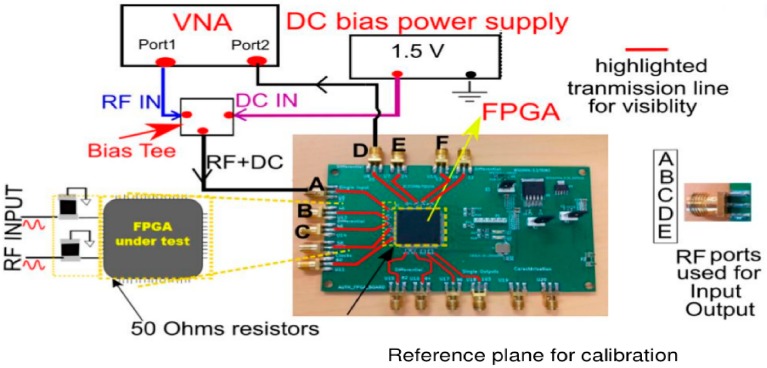
Measurement setup on the customized FPGA PCB to perform the RF test. Inset zoom shows an enlarge description of 50 Ohms resistors used with the transmission for proper matching. Input-Output (IO) ports naming is described which is used throughout this study. The reference plane for the calibration is also shown. VNA = vector network analyzer.

**Figure 5 sensors-20-02041-f005:**
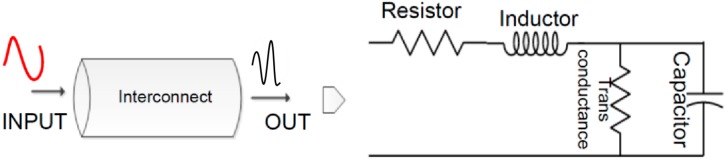
A resistor, inductor, capacitor (RLC) model of an interconnect. The model highlights the various passive elements that are present in a simple interconnect or wire, which effects the integrity of signals.

**Figure 6 sensors-20-02041-f006:**
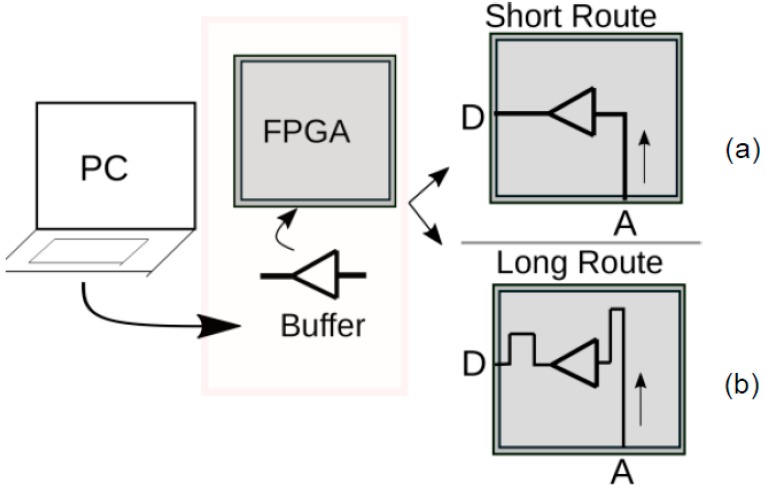
An illustration of programming FPGA with a buffer circuit and implementation of a route to the input-output port/pin (A and D in this Figure) of FPGA. (**a**) Implementation of short route between ports A and D. (**b**) Implementation of long route between port A and D.

**Figure 7 sensors-20-02041-f007:**
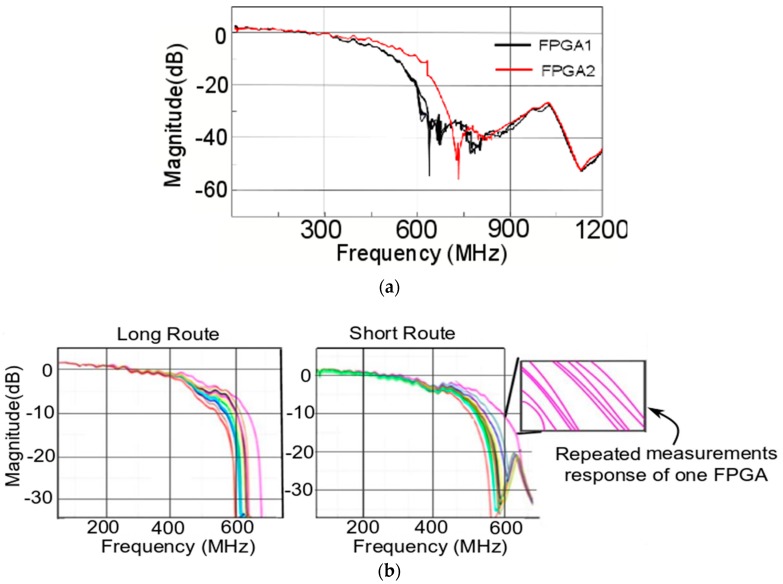
S21 response. (**a**) S21 response for full bandwidth measurement. (**b**) S21 response from the 11 FPGAs for two different routes, with each measurement done 10 times. Inset zoom on one of FPGA response to show the repeatability of measurement 10 times.

**Figure 8 sensors-20-02041-f008:**
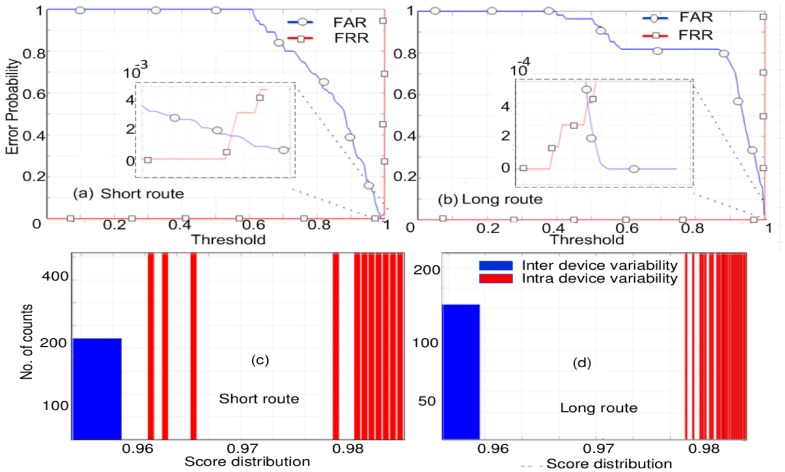
The error probability curves showing the distinction and overlap between false acceptance rate (FAR) and false rejection rate (FRR) with inset zoom on overlap of FAR and FRR. (**a**) Error probability curve for short route. (**b**) Error probability curve for long route. (**c**) Histogram describing the inter and intra variability for short route. (**d**) Histogram showing intra and inter variability for long route.

**Figure 9 sensors-20-02041-f009:**
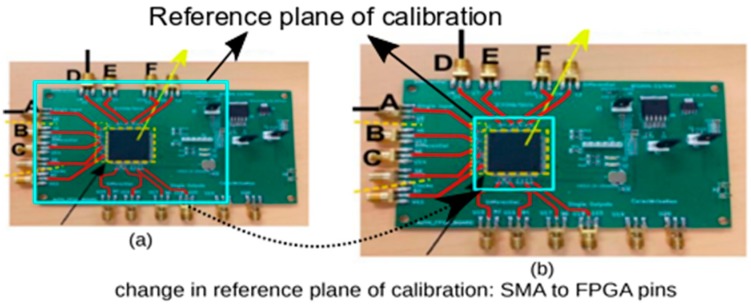
Changing reference plane of calibration. (**a**) Technique using difference of two routes to remove the systematic error relating to the PCB board. (**b**) Description of implementing the difference approach on the response of two routes of the FPGA.

**Figure 10 sensors-20-02041-f010:**
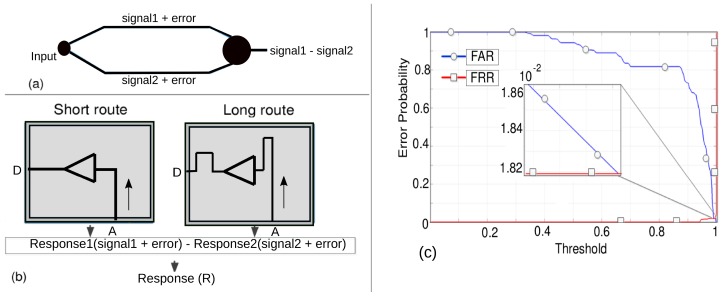
Error removal techniques. (**a**) Technique using difference of two routes to remove the systematic error. (**b**) Description of implementing the difference approach on the response of two routes of the FPGA. (**c**) Error probability curve after removal of the systematic error.

**Table 1 sensors-20-02041-t001:** Error probability for different cases.

Route Type	Long	Short	Difference of Routes (Error Removal)
Error Probability	10^−4^	10^−3^	10^−2^

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
