# Peer review of "Guided Electromagnetic Wave Technique for IC Authentication"

_sensors, 2020, doi:10.3390/s20072041_

Round 1

Reviewer 1 Report

  1. In Fig.7. It looks like resonance. Could the author plot the curves in more frequency band and deeper magnitude?
  2. If the complete curves in Fig.7 show resonance, it would be more due to with package instead of the chip itself. Could the authors give some explanation?

Reviewer 2 Report

This paper suggests a data-driven approach for IC authentication using microwaves. This is an interesting topic in emerging security applications from a hardware development perspective, as well as the machine learning-based processing approach.

  • It appears by only using RF frequencies below 1 GHz, this measurement is simply measuring the effective impedance of the whole IC, rather than providing local details. Acquiring detailed information about big ICs is necessary in security applications. I believe the frequency needs to be increased.
  • The authors have performed measurements with 10 dBm incident power, which generate relatively large EM fields (~1V / L_IC). This could cause interference issues in authenticating ICs under operation, because digital signals operate in the GHz range, same frequency as the RF used in this dielectric sensing work. Perhaps moving to higher frequencies could alleviate this issue.

Reviewer 3 Report

The proposed work basically utilizes the different frequency response, in this case S21 shown in Fig. 1, to distinguish different circuits. However, the situation may be oversimplified since the frequency response can be affected by a lot of factors such as bias voltage, temperature, and etc. In addition, the measurement has used 11 FPGAs to show the feasibility; nevertheless, when the volume of unknown circuits further increases, it may become more and more difficult to have enough margin for differentiation. 

Round 2

Reviewer 1 Report

As the authors said, the measured curves reflected the low pass response of the buffer circuits. So, the fig.7 could be updated to show the whole pictures like those in the cover letter. Only the rolling-down part of the response can tell the differences among the FPGA chips.
